# Weighted Gene Co-Expression Network Analysis Uncovers Core Drought Responsive Genes in Pecan (*Carya illinoinensis*)

**DOI:** 10.3390/plants14060833

**Published:** 2025-03-07

**Authors:** Mengxin Hou, Yongrong Li, Jiping Xuan, Yan Zhang, Tao Wang, Min Zhai, Guoming Wang, Longjiao Hu, Zhenghai Mo

**Affiliations:** 1Institute of Botany, Jiangsu Province and Chinese Academy of Sciences, Nanjing 210014, China; 2Nanjing Green Universe Pecan Science and Technology Co., Ltd., Nanjing 210007, China; 3Suqian Green Universe Pecan Science and Technology Co., Ltd., Suqian 223900, China; 4Jiangsu Engineering Research Center for the Germplasm Innovation and Utilization of Pecan, Nanjing 210014, China; 5Jiangsu Key Laboratory for the Research and Utilization of Plant Resources, Nanjing 210014, China

**Keywords:** drought, weighted gene co-expression network analysis, pecan, core genes

## Abstract

Drought severely affects the growth and production of pecan (*Carya illinoinensis*), while genes conferred drought adaptation are yet to be fully elucidated. Here, an in-depth exploration of the two different RNA-seq projects regarding drought stress (designated as P1 and P2) was performed via weighted gene co-expression network analysis. For the two projects, there existed one pair of modules (P1 turquoise module and P2 blue module) that was probably associated with drought resistance, as the paired modules both exhibited an increased expression profile with increasing water shortage stress and were annotated to be involved in oxidative stress response and the signaling pathways of abscisic acid and jasmonic acid. There were 441 and 1258 hub genes in the P1 turquoise module and P2 blue module, respectively, among which, 140 were overlapped and thus were recognized as core drought responsive genes. An additional drought stress experiment was conducted for RT-qPCR validation, and the results showed that the 20 core genes selected for detection were highly responsive to water deficit. Together, our results will be helpful for understanding the molecular mechanism of drought response and improving drought resistance in pecan.

## 1. Introduction

Drought is recognized as one of the future threats to our society because of global warming and climate change [1]. To overcome drought stress, plants have evolved various strategies, such as lowering stomatal aperture to control water loss, producing large and deep root systems to absorb water, accumulating kinds of solutes to reduce osmotic potential, and inducing antioxidant defense systems to maintain cellular reactive oxygen species (ROS) equilibrium [2,3,4]. The response of plants to water deficit involves numerous regulatory and functional proteins, functioning in various cellular processes [5]. The regulatory proteins, which can be grouped into three types, including transcription factors, protein kinases, and protein phosphatases, control the transduction of stress signals and the expression of functional genes [6]. Typical drought-responsive transcription factors include MYB, MYC, bZIP, CBFI, EREBP/APZ, and DREB1A [6]. The protein kinases are associated with the sensing and transduction of stress signals, such as mitogen-activated protein kinases (MAPKs), calcium-dependent protein kinases (CDPKs), and calcineurin B-like interacting protein kinase (CIPK) [7]. As for the protein phosphatases, they are related to second messenger production and enzyme transduction, such as phospholipase D and phospholipase C [8]. The functional genes for drought tolerance are mainly involved in osmotic adjustment, protein turnover, and ROS scavenging, such as *pyrroline-5-carboxylate reductase* (*P5CR*, associated with proline synthesis), *ABA-aldehyde oxidase* (*AAO*, participated in ABA synthesis), and *superoxide dismutase* (*SOD*) [9].

Pecan (*Carya illinoinensis*) is a nut tree that belongs to the Juglandaceae family. It is native to North America and has been widely cultivated in about 10 countries from all the continents except Antarctica [10,11]. Pecan nuts are not only delicious but also possess nutraceutical properties, since they contain mono-unsaturated fatty acids, phenolic compounds, zinc, and manganese that are beneficial for heart and brain function [12,13]. Generally, pecan is grown for its highly nutritious nuts; in some cases, it is planted for its wood, which is tough and can be used for flooring, veneer, tool handles, among other products [10]. In addition, pecan has landscape ecology applications, such as urban greening and beautification, environmental protection, and soil erosion prevention. The highly valued property of pecan has made its cultivated area grow upward [14]. The growth of pecan is frequently hindered by several abiotic stresses, such as waterlogging, drought, and spring freeze [15,16], among which the most prominent is drought. Drought could adversely impact the tree vigor and nut yield.

Pecan is somewhat resistant to waterlogging but is extremely sensitive to drought stress [17]. It has a large requirement for water, ranging from 600 to 1000 L per mature tree per day [18]. If drought stress appears, it decreases nut production, and the degree of influence relies on its timing, duration, and intensity [19]. A high-intensity or long-term water shortage could result in nut abortion and shoot growth cessation, dramatically reducing the final nut yield. A 4-year water stress experiment revealed that pecan yield would decrease by 24% when only 48% of the standard irrigation amount was supplied [20]. As for short-duration water deficit, it would lead to abnormal nut development. Pecan nut development can be divided into water, milk, dough, and kernel stages [21]. Water deficit during the nut water stage can decrease nut size, in the milk and dough stages reduce nut filling, and in the kernel stage delay shuck split [22]. For the young pecan trees, soil desiccation limits tree growth and even leads plants to mortality. As drought stress decreases pecan yield and retards growth, irrigation has become a common practice in pecan orchards to ensure an ample supply of water [23]. This is effective to mitigate water scarcity; however, genetic enhancement of drought tolerance is the most economical way. Understanding the genetic information regarding drought resistance will help in developing elite cultivars.

Nowadays, the genetic information for drought adaptation in pecan is still limited. There are two independent studies that have addressed the transcriptional responses of pecan to drought stress [17,24], and there may be common genes that are induced to alleviate this adverse condition. The shared genes obtained from different studies could represent the core drought-responsive genes. To gain a better understanding of the molecular mechanism concerning drought adaptation in pecan, we here integrated the data from different studies via a systems biology analysis method, weighted gene co-expression network analysis (WGCNA) [25]. Thousands of drought-responsive genes were grouped into modules, and then functional enrichment analysis was conducted to identify the shared stress-related pathways. The hub genes for the modules where those common pathways located were further detected to generate core drought-responsive genes.

## 2. Results

### 2.1. DEGs for Pecan in Response to Drought Stress

Two independent projects (P1 and P2) regarding drought stress were used for analysis in this study: P1 included dehydration and rehydration treatments, while P2 only contained the drought induction period (Figure 1a). The RNA-seq raw data were processed for gene expression level calculation, and the expression values were further used for DEG identification. For both projects, the number of DEGs was increased with the drought treatment time. The DEG number in P1 was increased from 1772 (D6) to 4213 (D24) and then decreased to 2175 after rehydration (Appendix A). As for P2, DEG number was consistently enhanced, varying from 477 (T3) to 11237 (T15) (Appendix A). The total DEG numbers in P1 and P2 were 6208 and 12981, respectively, and there were 4592 common DEGs for the two projects (Figure 1b). The 4592 common DEGs could be grouped into six clusters (C1–C6) based on their expression patterns (Figure 1c). Approximately half of the C1 genes in P1 were down regulated, while the rest were upregulated under dehydration stress. As for the C1 genes in P2, they were generally upregulated. The expressions for C2, C4, and C6 genes in the two projects were basically upregulated at certain time points after drought stress. On the whole, the C5 gene expressions in both projects were down regulated after drought treatment (without consideration for the rehydration period in P1). The C3 gene expressions were basically different between the two projects, exhibiting down-regulation in P1 but up-regulation in P2. The similar expressions for partially overlapped DEGs in the two projects suggested the existence of genes stably responsive to water deficit.

### 2.2. Co-Expression Networks Based on Drought-Related RNA-Seq

Selection of a suitable soft threshold (β) is the prerequisite for co-expression network construction. Our results showed that as the β enhanced, R^2^ was increased, while mean connectivity was decreased for both projects (Appendix A). When the R^2^ > 0.80 appeared for the first time in P1, the corresponding β was 24 (Appendix A). Since the highest R^2^ in P2 was 0.72, we selected R^2^ > 0.70 as the threshold, and in this situation, β = 24 was selected (Appendix A). Therefore, for the two projects, the suitable β was 24, and this value was proposed to build the co-expression network.

According to the hierarchical clustering (Figure 2a), the 4555 genes in P1 could be grouped into eight co-expression networks (modules), including black (178 genes), blue (838 genes), brown (522 genes), green (364 genes), red (326 genes), turquoise (1885 genes), yellow (418 genes), and gray (24 genes). As for P2, the total 8329 genes used for network construction were classified into nine modules, consisting of black (151 genes), blue (3128 genes), brown (421 genes), green (248 genes), pink (124 genes), red (157 genes), turquoise (3697 genes), yellow (358 genes), and gray (45 genes) (Figure 2b). Since the gray module was the representative of a set of genes that could not be assigned to any proper module, it was excluded from further analysis.

### 2.3. Expression Profile and Functional Enrichment for the Drought-Related Modules

To obtain an overview regarding the modules, expression dynamics and GO enrichments were conducted. The outcomes of the seven modules in P1 are shown in Figure 3. On the whole, genes in the blue and turquoise modules were gradually decreased and increased during the water deficit period (D6–D24), respectively. Therefore, the processes in the blue module, such as photosynthesis, photorespiration, starch biosynthetic process, and stomatal movement, may be inhibited. In contrast, kinds of stress-related cellular processes in the turquoise module were possibly induced under water deficit conditions, including response to hydrogen peroxide, response to heat, regulation of defense response, jasmonic acid (JA) mediated signaling pathways, abscisic acid (ABA) activated signaling pathway, and many more. Basically, gene expressions in the green module were consistently decreased during the dehydration and rehydration periods (D0–RD), and this module was mainly involved in cellular response to phosphate starvation. The black module, where genes were up regulated during the whole drought duration (D6–D24), was associated with carbohydrate transport. The red module exhibited a gradually decreased expression pattern from D6 to D24 but then increased on RD, and the enriched processes in this module included starch metabolic process and translational termination. For the brown and yellow modules, gene expressions were up regulated on specific time points after drought stress treatment.

The expression profile and GO enrichment result for the eight modules in P2 are present in Figure 4. In general, gene expressions in the blue and turquoise modules were gradually increased and decreased over time, respectively. The enriched biological processes in the blue module were mainly associated with stress response, including response to chitin, regulation of defense response, response to wounding, abscisic acid-activated signaling pathway, and regulation of JA-mediated signaling pathway, and many more. As for the turquoise module, light response-related processes, such as photosynthesis, chlorophyll biosynthesis, and photorespiration, were enriched. The other six modules, including black, brown, green, pink, red, and yellow, were up regulated on certain time points after drought stress. Among these, the black module seemed to be directly associated with drought resistance, as biological processes concerning L-proline biosynthesis and response to osmotic stress were enriched.

### 2.4. Potential Core Drought-Responsive Genes

Core-drought responsive genes should present similar expression patterns and functions under different water deficit conditions. For the two projects, the turquoise module in P1 and the blue module in P2 both displayed a gradually increased profile after dehydration and were simultaneously involved in stress response as revealed by the functional enrichment analysis. Therefore, the common hub genes in these two modules were considered as potential core-drought responsive genes. There were 441 and 1258 hub genes in the turquoise module of P1 and the blue module of P2, respectively (Figure 5a). Among them, 140 hub genes were overlapped for the two modules wherein gene expressions both peaked at the later stage of water deficit stress (Figure 5b). Out of the 140 common genes, 12 were annotated as transcription factor, including 4 MYB (Arabidopsis closest homolog: MYB3, MYB15, MYB108, and DIV2), 2 WRKY (WRKY48 and WRKY75), 2 trihelix (SLK2INT1 and GT-3a), 1 TCP (TCP9), 1 bZIP (bZIP38), 1 NF-YC (NF-YC2), and 1 NAC (NAC055) (Appendix A).

### 2.5. Validation of the Core Drought-Responsive Genes via RT-qPCR

To further verify the identified core drought-responsive genes, an additional water deficit treatment was conducted, and RT-qPCR was used to monitor the expression dynamics of 20 randomly selected genes. Results showed that all the detected genes were greatly induced under water shortage conditions, and all of them except *CiRLK* exhibited a more than 10-fold change (Figure 6). Consistent with the P1 and P2 projects, most of the genes (17 out of 20) displayed the peak expression levels at the later stage of water loss (after 24 h of treatment). Altogether, the core drought-responsive genes identified here were credible.

## 3. Discussion

In the present study, we used two RNA-seq projects to disclose the core drought responsive genes in pecan. Both projects indicated that the number of DEGs increased with increasing drought intensity (Appendix A), which is consistent with previous findings in maize (*Zea mays*) and *Abelmoschus esculentu* [26,27]. This tendency suggests that the higher the degree of water shortage, the more conspicuous the effect on gene transcription. Among the 4592 overlapping DEGs, those that shared similar expression profiles over the two studies were likely the key genes for drought adaptation. However, it is difficult to conclude which genes are more important based on cluster analysis, as this method is ineffective for dissecting the interconnection among genes [28]. To reveal the core drought responsive genes, we used WGCNA, which is a system biology method that could be effective for revealing gene–gene interaction patterns because of the network’s scale-free nature [25].

WGCNA groups thousands of genes into different modules, and functional enrichment analysis of the co-expressed gene modules can reveal the plant’s adaptation strategy to stress [29,30], which is the pre-requisite for identifying core drought-responsive genes for pecan. The two projects both indicated that the decrease in stomatal aperture and the activation of antioxidant defense systems were likely the primary adaptive mechanisms for pecan in response to drought stress, similar to the reports on maize [26], *Atractylodes lancea* [31], and *Chenopodium quinoa* [32]. Although this deduction lacks a direct experimental validation, it [26] can be inferred from the fact that the photosynthesis-associated genes were gradually decreased and the stress responsiveness-related genes were progressively increased with increasing water stress (Figure 4 and Figure 5). Closing stomata is a well-known strategy for drought avoidance, providing effective support for water conservation but directly influencing the photosynthetic efficiency and the carbon assimilation rate [4,33]. Therefore, a progressively decreased expression profile for photosynthesis-related genes might indicate a decrease in stomatal aperture. The antioxidant defense system is an important drought tolerance strategy for ROS clearance [34]. As water shortage stress increased, ROS was continuously accumulated, and coordinately, multiple biological processes (such as response to stress, oxidation-reduction, and protein folding) were induced to rescue ROS over-accumulation [35], since a high dose of which would ultimately lead to cell death [36]. In our study, stress-related processes including ‘response to hydrogen peroxide’, ‘response to wounding’, and ‘regulation of defense response’ were enriched, suggesting that the detoxification system was likely activated.

In addition to the direct involvement of the oxidative stress response, the turquoise module of P1 and the blue module of P2 were both involved in ABA and JA signal transductions. The ABA and JA hormones have been proven to possess multiple effects under drought stress, including the promotion of stomatal closure and the activation of the antioxidant defense system [37]. The regulation of the stomatal closure is governed not only by ABA, whose effect is principal, but also by JA. The two hormones could both induce the elevation of cytoplasmic calcium ion (Ca^2+^) in guard cells, and the Ca^2+^ is an essential secondary messenger in inducing stomatal closure [38]. The antioxidant defense-related role for ABA may be partially ascribed to its ability to promote the expression of various antioxidant genes, including *peroxidase*, *catalase*, *superoxide dismutase*, *ascorbate peroxidase*, *glutathione reductase*, and many more [39]. As for JA, the antioxidant role was owing to its capability in regulating the ascorbate–glutathione (GSH) cycle and polyphenol synthesis [40,41]. Exogenous application of ABA and JA has been reported to enhance the plant’s antioxidant defense system under drought stress [42,43]. In our study, ABA and JA signaling pathways occurred in the same module for both projects, suggesting that these two hormones might act synergistically under drought stress in pecan [44]. This finding is similar to the reports in *Oryza sativa* [45], *Triticum aestivum* [46], and *Salvia miltiorrhiza* [47]. Considering the primary adaptative mechanisms for pecan and the pleiotropy of ABA and JA, the turquoise module of P1 and the blue module of P2 were considered the key gene sets for drought response, and thus the 140 common hub genes were the core genes for drought response. Among them, the 20 core drought responsive genes selected for validation both showed highly induced (generally a 10-fold up regulation) by water deficit stress, suggesting that our results were highly reliable and may be applied in future breeding programs. The core drought-responsive genes obtained here were only identified from leaves. In fact, roots are also critical for mediating drought resistance via various physiological processes, such as water uptake, hormone synthesis, and respiration [48]. Further exploration using root tissue is essential to give a more comprehensive view of drought responsive genes.

The transcription factor can coordinately regulate the expression of kinds of down-stream genes [49], and thus the 12 transcription factors identified here may be the dominant controllers in the gene network. For those core transcription factors, their pivotal roles in mitigating drought stress have been reported partially in other plant species. In Arabidopsis, *AtMYB15* was capable of conferring drought tolerance via ABA-mediated stomatal closure [50]. Arabidopsis *AtMYB108* was implicated in both abiotic and biotic stress response [51], and its orthologue in *Paeonia lactiflora*, which was designated as *PlMYB108*, could increase drought resistance by enhancing the accumulation of flavonoid and the capability of ROS scavenging [52]. In *Populus wulianensis*, overexpression of *PwuWRKY48* (a *AtWRKY48* orthologue) was found to alleviate drought stress by increasing the ROS scavenging capacity and the proline content [53]. In *P. trichocarpa*, overexpressing *PtrWRKY75* (close to *AtWRKY75*) enhanced drought tolerance via salicylic acid-induced stomatal closure by ROS accumulation [54]. Arabidopsis AtbZIP38 (also named AtAREB2) was a master transcription factor in ABA signal transduction, and its knockout mutant showed an increased sensitivity to drought [55]. The overexpression of a conifer (*Picea wilsonii*) *PwHAP5* (Arabidopsis *NF*-*YC2* homolog) in *nf*-*yc2* Arabidopsis mutant has been proven to partially rescue the sensitivity to drought [56]. *RtNAC055*, an *NAC* gene from *Reaumuria trigyna* and close to Arabidopsis *AtNAC055*, could regulate the stomatal closure for drought resistance by sustaining the oxidant–antioxidant balance [57]. Those well documented genes further suggested the reliability for the core drought-responsive genes identified here.

## 4. Materials and Methods

### 4.1. RNA-Seq Data Processing

Two RNA-seq projects regarding pecan in response to drought stress were retrieved from NCBI (National Center for Biotechnology Information) under accession nos. PRJNA870326 and PRJNA799663. The PRJNA870326 project (P1) subjected seed-grown seedlings (four-months-old) to water deficit via withholding water continuously for 24 days (d). After 24 d of treatment, rehydration (RD) was performed, lasting 6 days. The leaves were collected at 0 (D0), 6 (D6), 18 (D18), 24 d (D24), and the RD period [17]. The PRJNA799663 project (P2) placed one-year-old grafted seedlings (‘Pawnee’ cultivar) to drought stress by persistently withholding water for 15 d, and the leaves were sampled at 0 (CK), 3 (T3), 6 (T6), 9 (T9), 12 (T12), and 15 d (T15) [24]. RNA-seq raw data were filtered by trimming adapter sequence, eliminating poor-quality and poly-N reads. Clean reads were thus obtained and further mapped to the reference genome of pecan cultivar ‘Pawnee’ [58]. The total counts for all the coding genes were computed and the gene expression levels were normalized to transcripts per million (TPM).

### 4.2. Identification of Differentially Expressed Genes (DEGs)

For both projects, gene expressions at 0 d were used as controls to detect DEGs. There were four comparisons in P1, including D6 vs. D0, D18 vs. D0, D24 vs. D0, and RD vs. D0. A total of five comparisons were performed for P2, including T3 vs. CK, T6 vs. CK, T9 vs. CK, T12 vs. CK, and T15 vs. CK. DEGs were detected via an R package DESeq2, and the criteria for differential expression were as follows: log2-fold change (FC) > 1 and false discovery rate (FDR) < 0.01.

### 4.3. Construction of Co-Expression Network

Genes with great variation (two-fold changes at least at one time point after drought stress) and high abundance (mean TPM values ranking in the top 60%; mean TPM > 3 for P1, and mean TPM > 9 for P2) were used for WGCNA. According to these criteria, the inputs for WGCNA comprised 4555 and 8329 genes in P1 and P2, respectively (Appendix A). Construction of co-expression network was conducted by an R package WGCNA. Firstly, the Pearson’s correlation coefficients were calculated for all the gene pairs using the signed parameter. After computation, a co-expression similarity matrix was generated. Secondly, a soft threshold (power, β) ranging from 1 to 30 was used to convert the similarity matrix into the adjacency matrix. When the scale-free topology fit index (R^2^) was larger than 0.80 for P1 and 0.70 for P2, the corresponding β was supposed to be suitable. Thirdly, the adjacency matrix was transformed into the topological overlap matrix (TOM) based on β = 24 and signed parameter. Finally, genes were divided into multiple co-expression gene sets (modules) according to the TOM-based dissimilarity. The least gene number in each module was set to 100, and the highly similar modules were merged based on the merge cut height of 0.25.

### 4.4. Functional Enrichment Analysis of Modules

Genes in the modules were extracted for functional annotation. An R package clusterProfiler was used for gene ontology (GO) enrichment analysis, and only the ontology of biological process was conducted. A GO term was considered as significantly enriched only when the corresponding adjusted *P* (*Padj*) values were smaller than 0.05. The outcomes of enrichment analysis as well as the expression pattern for each module were visualized by an R package ClusterGVis.

### 4.5. Selection of Hub Genes

The signed KME function embedded in the WGCNA package was applied to extract the eigengene-based connectivity (kME), which represents the Pearson relevance between the expression profiling of a certain gene and the general expression pattern of module genes. Genes with kME > 0.9 were recognized as hub genes.

### 4.6. Drought Stress Treatment

Fresh healthy seeds were collected from the trees of ‘Pawnee’ cultivar in September 2024 at our experimental farm. The harvested seeds were immersed in water for three days before subjecting to 30 °C incubator for germination. Two-month-old seedlings with about three true leaves were used for drought stress treatment, which was conducted according to a previous report [59]. Briefly, the leaves of the seedlings were detached and exposed to air under 25 °C and 70% relative humidity conditions. Pre-experiment showed that the leaves became wilted after 24 h of treatment. To explore the expressions patterns of drought related genes, leaves were collected at 0, 6, 12, and 24 h after water loss. Three biological repetitions were conducted for each time point, and the samples for each repetition were collected from three individual plants. The samples were quickly frozen in liquid nitrogen before placing to a −80 °C refrigerator.

### 4.7. Validation the Responsiveness of Drought Related Genes via RT-qPCR

Total RNA was extracted via a Universal Plant RNA kit (BioTeke, Beijing, China) according to the guideline instructions. The integrity and concentration of isolated RNA were determined by 1% agarose gel electrophoresis and a Nanodrop spectrophotometer (Thermo Fisher Scientific, Wilmington, DE, USA), respectively. Only non-degraded RNA that had an A260/A280 ratio of 1.8–2.2 and an A260/A230 ratio larger than 2.0 were kept for further processing. DNA elimination and first-strand cDNA synthesis were conducted using the PrimeScript™ RT Reagent kit with gDNA Eraser (TaKaRa, Kyoto, Japan) according to the manufacture`s protocol. A total of 20 core drought-related genes, which have been proven to be associated with drought stress in other species, were selected for verification of their responses to water deficit. Primers were designed using Beacon Designer v8 software with default parameters, and all the primers were supplied in Appendix A. Gene expression was determined on a CFX Opus 96 Real-Time PCR System (Bio-Rad, Hercules, CA, USA) with TB Green Ex Taq™ II kit (TaKaRa, Kyoto, Japan). The qPCR procedure was as follows: DNA denaturation at 95 °C for 3 min; 40 cycles of amplification at 95 °C for 10 s and 60 °C for 30 s; amplicon melting analysis by rising the temperature from 65 °C to 95 °C by 0.2 °C at each step. The expression value was calculated using 2^−ΔΔCt^ method with actin for normalization.

## 5. Conclusions

In our study, two different RNA-seq projects regarding drought stress were used to identify core drought responsive genes. WGCNA grouped the variable genes into different modules, among which there existed a module for each project showing a gradually increased expression pattern with increasing drought stress. The two modules were both directly and indirectly involved in stress adaptation. A total of 140 hub genes were overlapped in the two modules, and these genes were considered as core drought responsive genes in pecan. A total of 20 core drought-responsive genes were chosen for further validation in an additional water-deficit treatment, and RT-qPCR detection confirmed that their expressions were highly induced by drought stress. The 140 drought responsive genes will serve as valuable resources for future research and breeding. Selecting the core genes to validate their biological functions may be helpful for better understanding the molecular mechanism of drought adaptation in pecan. Utilizing the core drought responsive genes in breeding via molecular breeding tools, such as molecular marker selection and gene editing, will make it possible to develop new pecan cultivars with desired drought resistance trait.

## Figures and Tables

**Figure 1 plants-14-00833-f001:**
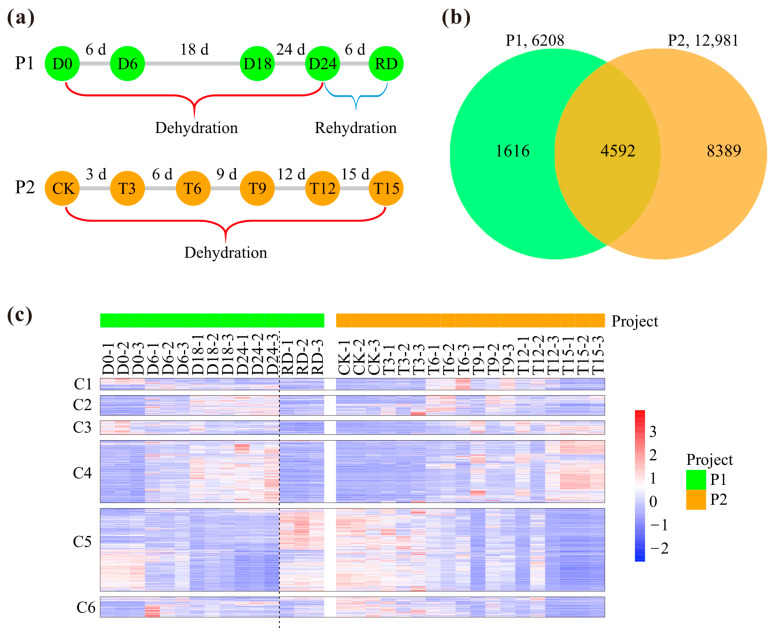
The number and expression patterns of differentially expressed genes (DEGs) for pecan in response to drought stress derived from two different projects (P1 and P2). (**a**) Experimental scheme depicting the drought treatment in P1 and P2. P1 includes the dehydration (D) and rehydration (RD) periods, and P2 contains only water-deficit treatment (T). (**b**) The number of DEGs for P1 and P2. (**c**) Expression patterns for the 4592 common DEGs are grouped into six clusters (C1–C6). Expression levels for each project are independently z-score normalized.

**Figure 2 plants-14-00833-f002:**
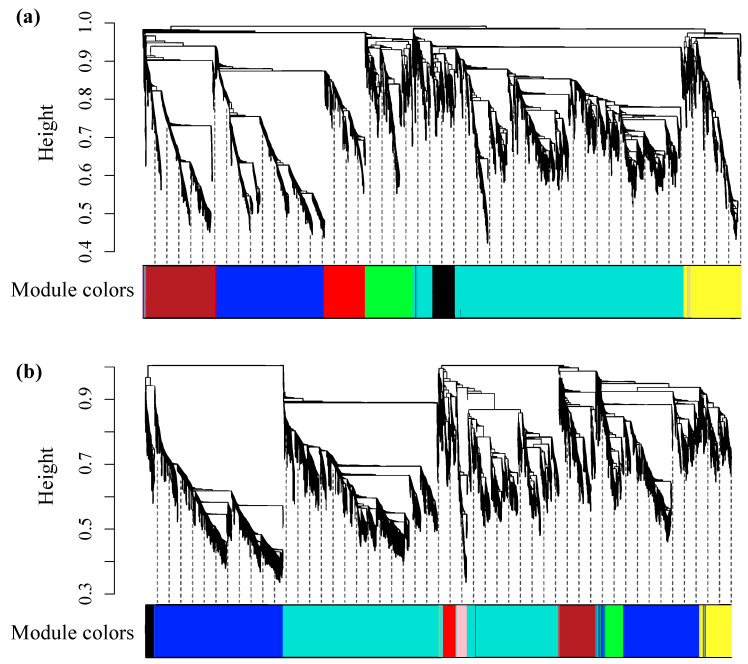
Hierarchical clustering of the genes used for co-expression network construction. (**a**) Genes clustering in project 1 (P1). (**b**) Genes clustering in P2. For the clustering, y-axis suggests expression dissimilarity between neighboring genes, and each branch indicates one gene. Module colors represent the module designation for the co-expressed genes.

**Figure 3 plants-14-00833-f003:**
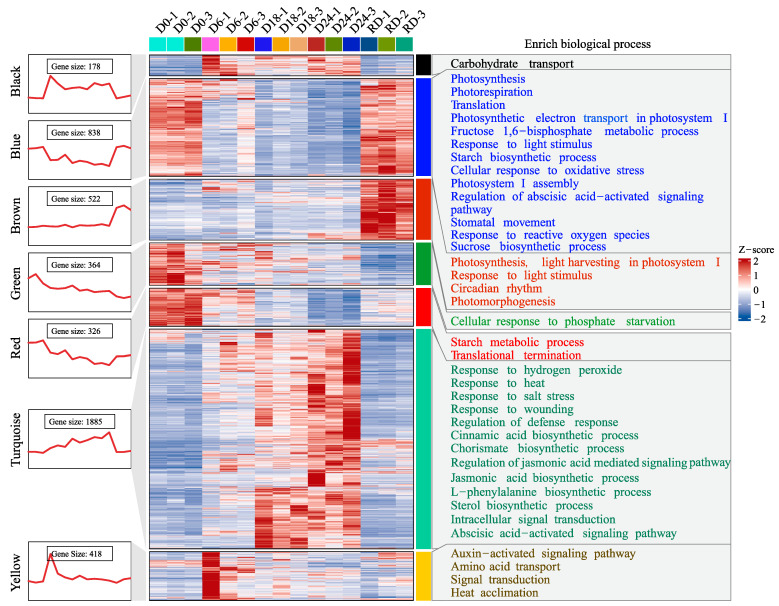
Expression pattern and gene ontology functional enrichment for the modules in project 1. Gene expressions are z-score normalized and visualized via line charts as well as heatmaps. D0–D24 represent the samples collected 0~24 days after dehydration. RD is the sample collected after rehydration. Then, -1, -2, and -3 are the three different biological replications.

**Figure 4 plants-14-00833-f004:**
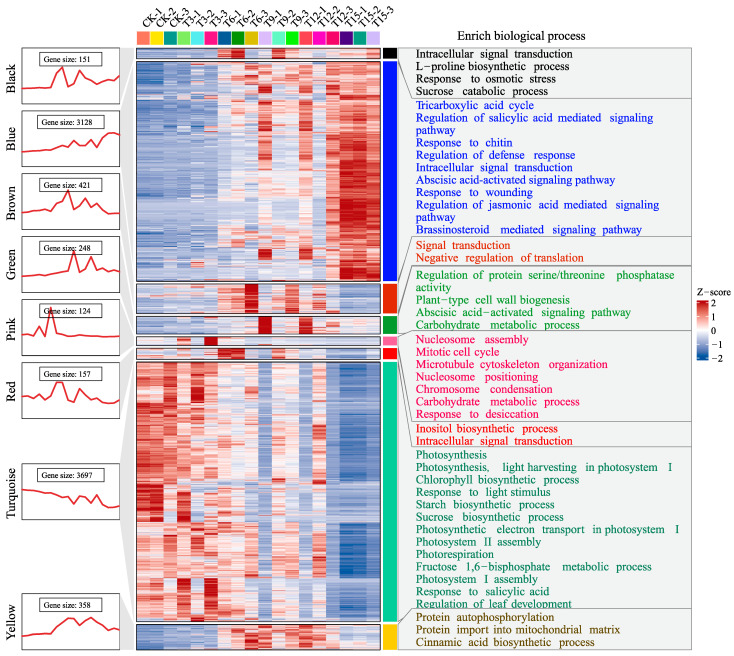
Expression pattern and gene ontology functional enrichment for the modules in project 2. Gene expressions are z-score normalized and visualized via line charts as well as heatmaps. CK–T15 represent the samples collected 0~15 days after drought stress. Then, -1, -2, and -3 are the three different biological replications.

**Figure 5 plants-14-00833-f005:**
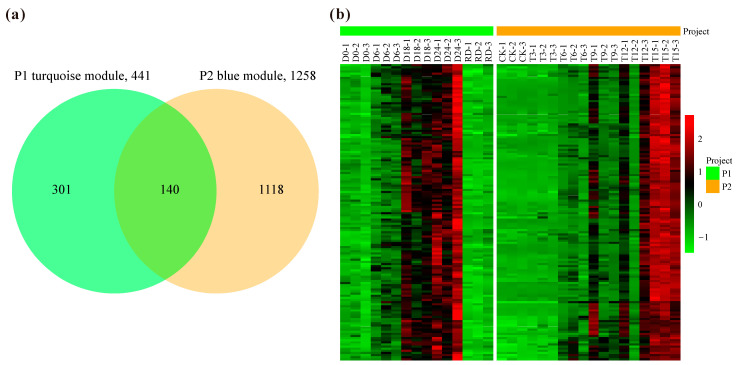
The number of hub genes (**a**) and the expression patterns for the common genes (**b**) in the turquoise module of project 1 (P1) and the blue module of P2. Expression patterns are z-scores independently for P1 and P2.

**Figure 6 plants-14-00833-f006:**
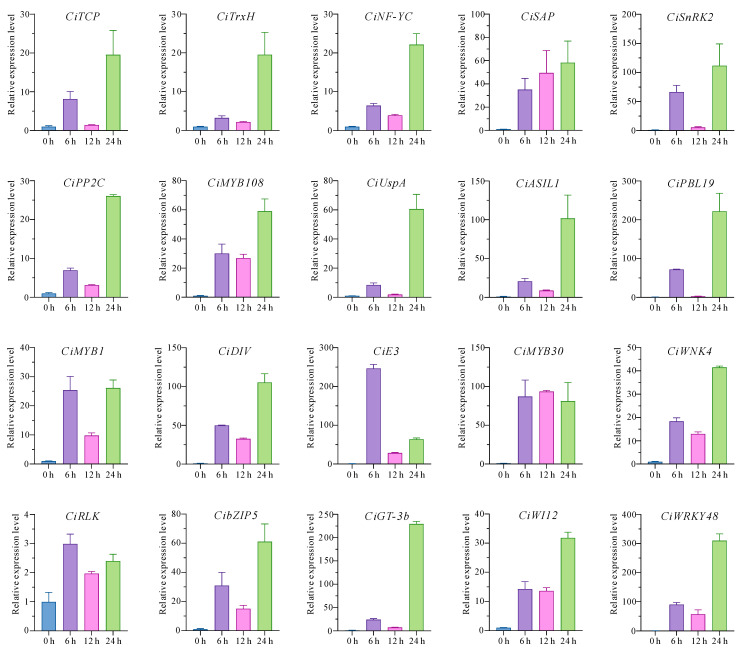
The expression patterns of 20 core-drought responsive genes detected via RT-qPCR. The detached leaves were sampled after 0, 6, 12, and 24 h of water loss.

## Data Availability

The data presented in this study are available in this article.

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
