# Peer review of "Weighted Gene Co-Expression Network Analysis Uncovers Core Drought Responsive Genes in Pecan (Carya illinoinensis)"

_plants, 2025, doi:10.3390/plants14060833_

Round 1

Reviewer 1 Report

Comments and Suggestions for Authors

Dear Dr. Hou et al., Your work on drought-responsive genes in pecan (Carya illinoinensis) is poised to be a valuable contribution to the field. The authors have employed a robust approach by integrating data from two independent RNA-seq projects to identify core drought-responsive genes in this economically important nut crop.

The study's primary strength lies in its use of weighted gene co-expression network analysis (WGCNA) to uncover key gene modules and hub genes associated with drought response. This systems biology approach provides a more comprehensive understanding of the complex molecular mechanisms underlying drought tolerance in pecan. The identification of 140 common hub genes, including 12 transcription factors, offers valuable targets for future functional studies and potential genetic improvement of drought tolerance in pecan.

The authors have also demonstrated thoroughness in their analysis by conducting an additional drought stress experiment and validating 20 randomly selected core genes using RT-qPCR. This validation step strengthens the reliability of their findings and provides confidence in the identified core drought-responsive genes.

However, there are several areas where the manuscript could be improved to enhance its impact and clarity. Firstly, although the authors mention the economic importance of pecan, they could strengthen the introduction by providing more specific data on the crop's global production and the estimated economic losses due to drought stress. This would better contextualize the significance of their research for both scientific and agricultural audiences.

The methods section, although generally well-described, could benefit from more detailed justification for certain analytical choices. For instance, the authors could explain why they chose specific thresholds for gene selection in the WGCNA analysis (mean TPM > 3 for P1 and mean TPM > 9 for P2). Additionally, the rationale behind selecting 20 genes for RT-qPCR validation could be elaborated upon. What is the cultivar for P1, and why was that chosen? What is its relathionship (if any) to the cultivar in P2, and how does that reflect the species' diversity?

In the results and discussion sections, the authors could enhance the manuscript by providing a more in-depth comparative analysis of their findings with similar studies in other crop species. This would help readers understand how the drought response mechanisms in pecan align with or differ from those in other plants, potentially revealing unique adaptations in this species. Was there any population genomics study done on your crop, to see whether your drought-responsive genes are under selection?

The manuscript would also benefit from a more detailed exploration of the potential applications of these findings. Although the authors mention that their results will be helpful for enhancing drought tolerance in pecan, they could expand on specific, tangible, testable strategies for utilizing this knowledge in breeding programs or genetic engineering approaches.

Furthermore, the discussion could be strengthened by addressing potential limitations and any biases of the study. For example, the authors could discuss how the controlled conditions of their experiments might differ from field conditions and how this could impact the applicability of their findings. Or how the findings in leaves may not be directly translatable to fruit/yield.

Lastly, the figures, though informative, could be improved for clarity. Some of the text in the figures (particularly in Figure 1) is quite small and may be difficult to read. 

With some revisions to enhance clarity, provide more comparative context, and discuss potential applications and limitations, this work has the potential to make importnant contributions to our understanding of drought tolerance in this important crop species.

Comments on the Quality of English Language

This manuscript should undergo copy-edit in a professional scientific writing office or using your professional network. Focus on clarity and accuracy. Some words (osmatic, other examples present) do not exist in English. Do not use 'significant' unless accompanied by robust statistical testing - other adjectives with that meaning exist. Similar for 'seriously' - as opposed to what, 'jokingly'?

Author Response

Comments 1: there are several areas where the manuscript could be improved to enhance its impact and clarity. Firstly, although the authors mention the economic importance of pecan, they could strengthen the introduction by providing more specific data on the crop's global production and the estimated economic losses due to drought stress. This would better contextualize the significance of their research for both scientific and agricultural audiences.

Responses 1: Thank you for pointing this out. There are few reports on the economic losses due to drought stress for pecan, since this tree has alternate bearing characteristic and numerous factors can affect yield, making it difficult to estimate the yield loss in production. We only find one study about the yield loss because of water deficit, we have cited it accordingly in the introduction section.

Comments 2: The methods section, although generally well-described, could benefit from more detailed justification for certain analytical choices. For instance, the authors could explain why they chose specific thresholds for gene selection in the WGCNA analysis (mean TPM > 3 for P1 and mean TPM > 9 for P2). Additionally, the rationale behind selecting 20 genes for RT-qPCR validation could be elaborated upon. What is the cultivar for P1, and why was that chosen? What is its relathionship (if any) to the cultivar in P2, and how does that reflect the species' diversity?

Responses 2: Thanks for your comment. The thresholds for gene selection during WGCNA and RT-qPCR have been supplied in the material and method section. For WGCNA, genes with mean TPM values ranking in the top 60% are selected (Line 338). For RT-qPCR, genes that have been reported to be related with drought stress are selected (Line 385). The materials for P1 are the seed-grown seedlings (Line 317). Pecan is highly heterozygous and thus the seedlings used in P1 do not belong to any cultivar. As for P2, grafted seedlings are used and the materials are the clone of ‘Pawnee’ cultivar (Line 321). These two projects used different genotypes and thus our study can reflect genotype and treatment diversity.

Comments 3: In the results and discussion sections, the authors could enhance the manuscript by providing a more in-depth comparative analysis of their findings with similar studies in other crop species. This would help readers understand how the drought response mechanisms in pecan align with or differ from those in other plants, potentially revealing unique adaptations in this species. Was there any population genomics study done on your crop, to see whether your drought-responsive genes are under selection?

Responses 3: Thank you for pointing this out. Our main findings include the drought adaption mechanism and the critical drought responsive genes. Since we aim to find the overlapped drought response between the two different projects, the drought adaptation we revealed are quite common, an no unique adaptations are identified. We have added other similar reports concerning the mechanism of drought adaptation (Line 247-248 and Line 276-279). As for the drought responsive genes, they have been discussed in detail in Line 291-312. There are no population genomics on the drought trait for pecan, so we fail to provide more information.

Comments 4: The manuscript would also benefit from a more detailed exploration of the potential applications of these findings. Although the authors mention that their results will be helpful for enhancing drought tolerance in pecan, they could expand on specific, tangible, testable strategies for utilizing this knowledge in breeding programs or genetic engineering approaches.

Responses 4: Agree. The potential application for our study has been provided in the abstract and conclusion section.

Comments 5: Furthermore, the discussion could be strengthened by addressing potential limitations and any biases of the study. For example, the authors could discuss how the controlled conditions of their experiments might differ from field conditions and how this could impact the applicability of their findings. Or how the findings in leaves may not be directly translatable to fruit/yield.

Responses 5: Thank you for pointing this out. In the new version, the limitations are discussed based on the results including the drought mechanism derived from deduction (Line 248) and the core genes confining to the leaf tissue (Line 286-290).

Comments 6: Lastly, the figures, though informative, could be improved for clarity. Some of the text in the figures (particularly in Figure 1) is quite small and may be difficult to read. 

Responses 6: Thank you for pointing this out. Figure 1 and Figure 5 has been modified to make the text more readable.

Comments 7: With some revisions to enhance clarity, provide more comparative context, and discuss potential applications and limitations, this work has the potential to make important contributions to our understanding of drought tolerance in this important crop species.

Responses 7: Thans for your comments. We have revised them accordingly.

Comments 8: This manuscript should undergo copy-edit in a professional scientific writing office or using your professional network. Focus on clarity and accuracy. Some words (osmatic, other examples present) do not exist in English. Do not use 'significant' unless accompanied by robust statistical testing - other adjectives with that meaning exist. Similar for 'seriously' - as opposed to what, 'jokingly'?

Responses 8: Thans for your comments. We have checked and revised our English writing thoroughly. We did a statistical testing for GO enrichment, so ‘significant’ appeared in the corresponding sentence is retained.

Reviewer 2 Report

Comments and Suggestions for Authors

Comments and Suggestions for Authors:
The work submitted by Hou et al., "Weighted gene co-expression network analysis uncovers core drought responsive genes in pecan (Carya illinoinensis)" reports an analysis of two RNA-seq datasets obtained from the NCBI via weighted gene co-expression network analysis. The two projects are related to expression analysis under water stress. While in the first project (P1) four month old seedlings were withheld from water for 24 days and then rehydrated continuously for six days, in the second project (P2) one year old grafted seedlings were subjected to water stress by withholding water for 15 days. In addition, the authors did a water stress experiment in conditions different from the ones used to obtain the datasets under study. The authors confirmed several genes found in the reported analysis, indicating a serious and well-structured gene co-expression network analysis.

In general, the authors did an excellent and methodic work, but they should consider making a few changes to improve the quality of the paper.

Abstract:
The abstract concisely describes the main results.

Minor changes in the wording to consider:
L20 remove separately

L21 add “respectively” after module

L22 In the sentence “An additional drought stress experiment was conducted and RT-qPCR further confirmed that the identified core genes were highly responsive to water deficit.” The authors only analyzed a set of genes by RT-PCR, not all of them.

1. Introduction
The introduction, while well-structured, needs to improve the flow and grammar.

Minor changes to consider:
L83 reference for weighted gene co-expression network analysis (WGCNA)

L84 remove were

2. Results
The findings are well-structured and organized, featuring well-described figures that enhance understanding. However, some changes might help readers better understand the manuscript.

Minor changes to consider:

2.1. DEGs for pecan in response to drought stress

The results will be more evident with a figure (it might be Figure 1a) that describes how the experiments for projects one and two were set. This change helps the reader quickly understand the analysis since the rehydration on project one can be confusing.

In Figure 1b P1, adding a vertical line that separates drought induction from dehydration might help the reader understand the figure better.

2.2. Co-expression networks based on drought-related RNA-seq

Figure 2 might not be necessary directly on the paper. It can be added as a supplementary figure for material and methods.

3. Discussion

The discussion is well-structured but does not detail the experimental confirmation of core gene differential regulation under drought stress. The author should expand on this part since it helps to point out the quality of the analysis. The expression pattern of these genes clearly indicates that the authors exposed a group of candidate genes that might be of great importance, for example, in genetic improvement approaches.

Some minor changes to consider:

L220 It might be precise to say over “these two studies” and not “multiple studies”

L225 “because of its unique algorithm” sounds a bit vague.

4. Materials and Methods

The authors clearly present the experimental procedure and correctly cite the two datasets used in this study.

4.3. Construction of co-expression network

The authors should consider describing Figure 2 in this section and add it as a supplementary figure.

5. Conclusions

The conclusions summarize the manuscript. However, there is no mention about the drought stress experiment produced by the authors and the differential expression confirmation of the candidates genes.

Minor changes to consider:

L361 “used” or a similar word might be more appropriate than “applied”

Comments on the Quality of English Language

In general, it is easier to read, but a few improvements can be made.

Author Response

Comments 1: The abstract concisely describes the main results. Minor changes in the wording to consider: L20 remove separately; L21 add “respectively” after module; L22 In the sentence “An additional drought stress experiment was conducted and RT-qPCR further confirmed that the identified core genes were highly responsive to water deficit.” The authors only analyzed a set of genes by RT-PCR, not all of them.

Responses 1: Thank you for pointing this out. We have revised them accordingly in the new version.

Comments 2: The introduction, while well-structured, needs to improve the flow and grammar. Minor changes to consider: L83 reference for weighted gene co-expression network analysis (WGCNA); L84 remove were

Responses 2: Thank you for pointing this out. We have modified them accordingly in the new version.

Comments 3: The findings are well-structured and organized, featuring well-described figures that enhance understanding. However, some changes might help readers better understand the manuscript. Minor changes to consider: 2.1. DEGs for pecan in response to drought stress; The results will be more evident with a figure (it might be Figure 1a) that describes how the experiments for projects one and two were set. This change helps the reader quickly understand the analysis since the rehydration on project one can be confusing. In Figure 1b P1, adding a vertical line that separates drought induction from dehydration might help the reader understand the figure better; Figure 2 might not be necessary directly on the paper. It can be added as a supplementary figure for material and methods.

Responses 3: Thanks very much for your insightful comments. In the new version, experimental scheme depicting the drought treatment in the two projects has been supplied in Figure 1a; A vertical line that separates dehydration from rehydration has been added in Figure 1c; The previous Figure 2 has been transferred to supplementary files.

Comments 4: The discussion is well-structured but does not detail the experimental confirmation of core gene differential regulation under drought stress. The author should expand on this part since it helps to point out the quality of the analysis. The expression pattern of these genes clearly indicates that the authors exposed a group of candidate genes that might be of great importance, for example, in genetic improvement approaches. Some minor changes to consider: L220 It might be precise to say over “these two studies” and not “multiple studies”; L225 “because of its unique algorithm” sounds a bit vague.

Responses 4: Thanks very much for your constructive comments. The verification of the core genes has been added in the discussion section (Line 283-285); ‘Over multiple studies’ had been revised to ‘over the two studies’’ (Line 236); ‘because of its unique algorithm’ is modified to ‘because of the network`s scale-free nature’ (Line 240).

Comments 5: The authors clearly present the experimental procedure and correctly cite the two datasets used in this study; The authors should consider describing Figure 2 in this section and add it as a supplementary figure.

Responses 5: Thanks for your comments. We have ensured that the RNA-seq raw data are correctly cited, and the previous Figure 2 has been transferred to supplementary file (Figure S2).

Comments 6: The conclusions summarize the manuscript. However, there is no mention about the drought stress experiment produced by the authors and the differential expression confirmation of the candidates genes. Minor changes to consider: L361 “used” or a similar word might be more appropriate than “applied”

Responses 6: Thank you for pointing this out. An additional water deficit experiment and RT-qPCR detection conducted by ourselves have been mentioned in the conclusion version (Line 401-403); The word ‘applied’ has been replaced by ‘used’ (Line 396).

Comments 7: Comments on the quality of English language. In general, it is easier to read, but a few improvements can be made.

Responses 7: Thanks for your comment. We have checked and revised our English writing thoroughly.

Reviewer 3 Report

Comments and Suggestions for Authors

The article "Weighted gene co-expression network analysis uncovers core drought responsive genes in pecan (Carya illinoinensis)" by Hou et al. is well written, the results obtained from the applied analyses and their interpretation are correct. The authors analyzed two previously annotated RNA-seq projects concerning drought stress to identify the core genes responding to drought in pecan. A total of 140 hub genes were identified as core genes responding to drought in pecan.

I believe the colors used to represent the data from the two projects - P1 and P2 - should be the same in all figures in the article. For example, Figure 1A - Project 1 is pink, and Project 2 is green, while Figure 1B is the other way around!

The discussion could be expanded with additional literature data in other species supporting the role of the proposed genes in drought-responsive pecan.

Author Response

Comments 1: I believe the colors used to represent the data from the two projects - P1 and P2 - should be the same in all figures in the article. For example, Figure 1A - Project 1 is pink, and Project 2 is green, while Figure 1B is the other way around!

Responses 1: Thank you for pointing this out. We have made the colors representing P1 and P2 be the same in all the figures.

Comments 2: The discussion could be expanded with additional literature data in other species supporting the role of the proposed genes in drought-responsive pecan.

Responses 2: Thanks for your comment. After carefully reviewing all the relevant literatures, the core gene about WRKY75 is discussed (Line 302-303) to support the reliability of our results.

Round 2

Reviewer 3 Report

Comments and Suggestions for Authors

I believe the article is suitable for publication in its present form in the Plants journal.